# Dynamics of Replication-Associated Protein Levels through the Cell Cycle

**DOI:** 10.3390/ijms25158230

**Published:** 2024-07-28

**Authors:** Aleksandar Atemin, Aneliya Ivanova, Petar-Bogomil Kanev, Sonya Uzunova, Marina Nedelcheva-Veleva, Stoyno Stoynov

**Affiliations:** Laboratory of Genomic Stability, Institute of Molecular Biology, Bulgarian Academy of Sciences, Acad. G., Bonchev Str. Bl. 21, 1113 Sofia, Bulgaria; atemin@bio21.bas.bg (A.A.); anivanova@bio21.bas.bg (A.I.); pkanev@bio21.bas.bg (P.-B.K.); sonyauzunova84@gmail.com (S.U.); marina@bio21.bas.bg (M.N.-V.)

**Keywords:** cell cycle, PCNA, RIF1, ORC1, MCM6, Claspin, live-cell imaging, spinning-disk confocal microscopy

## Abstract

The measurement of dynamic changes in protein level and localization throughout the cell cycle is of major relevance to studies of cellular processes tightly coordinated with the cycle, such as replication, transcription, DNA repair, and checkpoint control. Currently available methods include biochemical assays of cells in bulk following synchronization, which determine protein levels with poor temporal and no spatial resolution. Taking advantage of genetic engineering and live-cell microscopy, we performed time-lapse imaging of cells expressing fluorescently tagged proteins under the control of their endogenous regulatory elements in order to follow their levels throughout the cell cycle. We effectively discern between cell cycle phases and S subphases based on fluorescence intensity and distribution of co-expressed proliferating cell nuclear antigen (PCNA)-mCherry. This allowed us to precisely determine and compare the levels and distribution of multiple replication-associated factors, including Rap1-interacting factor 1 (RIF1), minichromosome maintenance complex component 6 (MCM6), origin recognition complex subunit 1 (ORC1, and Claspin, with high spatiotemporal resolution in HeLa Kyoto cells. Combining these data with available mass spectrometry-based measurements of protein concentrations reveals the changes in the concentration of these proteins throughout the cell cycle. Our approach provides a practical basis for a detailed interrogation of protein dynamics in the context of the cell cycle.

## 1. Introduction

Through the concerted regulation of gene expression, as well as protein localization and function, the cell cycle drives cellular growth and proliferation [1,2,3,4,5,6]. Cycle control is enacted through a sophisticated regulatory network, defined by cyclins and cyclin-dependent kinases (CDKs), whose function is generally dependent on mitogenic signals and favorable conditions to propel the cycle forward [3,4,5]. In opposition, an arsenal of anti-mitogenic signaling factors that constitute multiple checkpoints throughout the cycle is usually in place to halt cycling and trigger exit into G0 (quiescence) or even cell death in response to stresses, such as DNA damage and low nutrient availability [1]. The overall cycle length, as well as the length of specific phases, vary widely throughout development, in addition to between and within tissues [7,8]. Such variations are also observed among cells under identical conditions and thus complicate fluorescence microscopy-based quantification of specific protein levels throughout the cell cycle.

A deregulated cycle takes center stage in the process of tumorigenesis, where checkpoints are compromised, and mitogens may no longer be required, which gives rise to so-called unscheduled proliferation [9]. A deregulated cell cycle is accompanied by replication stress and genomic instability as well-established and interlinked hallmarks of malignancy [10]. The ever-greater efforts to decipher the molecular pathogenesis of cancer are defined by the functional characterization of various proteins involved in replication, transcription, and DNA repair. More often than not, these proteins are subject to cell cycle-dependent regulation for their expression and activity. At any given point, their levels are the net result of multiple factors, namely, gene expression, mRNA degradation, translation, and protein degradation [11]. Cyclins represent archetypal factors whose expression oscillates under cell cycle control, with different cyclins accumulating at different stages when their function is required to propel the cycle forward [12]. Beyond cyclins, a myriad of proteins exhibit cell cycle-dependent expression patterns. In fact, a considerable part of DNA repair and replication-associated factors oscillate throughout the cell cycle. For example, the levels of replication origin licensing factor ORC1 peak in G1 and are then degraded during S [13]. 

Interestingly, those of the other ORC subunits remain constant [14]. Homologous recombination factor RAD51, which acts from S phase onwards, continuously accumulates throughout the cycle, peaking in G2/M [15,16]. The levels of proliferating cell nuclear antigen (PCNA), the replication clamp, are scarce at the beginning of G1 and then peak in S phase, remaining abundant thereafter [17]. It should, however, be noted that protein levels alone are far from sufficient to recapitulate the engagement of these factors in their functional “calling”, which is often reflected by their accumulation into foci or other “structures” discernible through fluorescence microscopy (e.g., the accumulation of PCNA in replication foci during S-phase) [18,19].

The need for interrogating cycle-dependent protein dynamics in the context of fundamental as well as disease-oriented research has fostered the development of various methods, which can be broadly divided into bulk biochemical and fluorescence-based approaches [20]. Most of these allow for protein-specific analysis in the context of the cell cycle yet come with certain limitations. For example, while western blotting-based quantification can yield a phase-by-phase picture of specific protein levels, it requires the use of synchronization drugs, which in their own right affect cellular behavior (including protein turnover) [20]. The temporal resolution obtained through this method is poor, while virtually no spatial (subcellular) context is provided. Further, it cannot be applied at the single-cell level. Classical immunofluorescence-based approaches utilizing nucleotide analogs or replication-associated factors (e.g., PCNA) do provide insights into the spatial distribution at the single-cell level but are limited in temporal resolution due to the requirement for cell fixation [20]. 

In contrast, fluorescence-based reporters allow one to follow a protein of interest in real-time throughout the cell cycle, at the single-cell level, and with sufficient temporal resolution. The most widely used is the Fluorescence Ubiquitin Cell Cycle Indicator (FUCCI) technology [21], which is based on the reciprocal expression patterns of licensing factor Cdt1 and Geminin. FUCCI has been adapted for application in several model systems, including mice and zebrafish [21]. It is, however, limited by the utilization of synthetic promoters that drive higher than the physiological expression levels of reporter-tagged proteins, as well as by the need to introduce a third fluorescently tagged protein, that is, the protein of interest, which may require the use of fluorescent tags that are excited by phototoxic wavelengths of laser light (e.g., 405 nm). With regard to temporal resolution, while FUCCI enables researchers to follow cells throughout the cell cycle, it may not achieve the desired discrimination between phases and “subphases” (e.g., early, middle, and late S). Taken together, an optimized approach for the study of specific proteins in a cell cycle context without the above-described limitations is highly desired, even more so if it would allow for some measure/indicator of protein function/engagement. Observing PCNA subnuclear localization through microscopy has been proposed as a tool for the effective discrimination between cell cycle phases [22]. The distinct profiles of PCNA replication foci between the subphases of S represent a phenotype that can be exploited for distinguishing between these subphases automatically. However, mean fluorescence intensity data of PCNA in cells are insufficient for this purpose. Thus, approaches for intensity data processing that would reflect foci size and abundance, such as edge enhancement filters, hold promise for effective subphase discrimination based on PCNA [23].

Herein, we employ time-lapse live-cell imaging to follow and quantify the dynamic changes in protein levels during the cell cycle at single-cell resolution. We use transgenic cell lines co-expressing mCherry-tagged mouse PCNA (mPCNA) and an EGFP-tagged protein of interest, both under the control of their endogenous regulatory sequences. Converting PCNA signal intensity via Sobel filtering allowed us to effectively discriminate between cell cycle phases as well as between early, middle, and late S. Thus, we characterized in great spatiotemporal detail the dynamics of protein levels for multiple replication-associated factors, namely, origin recognition complex subunit 1 (ORC1), Rap1-interacting factor 1 (RIF1), Claspin, PCNA, and minichrosome maintenance complex component 6 (MCM6). 

## 2. Results and Discussion

### 2.1. Double-Tagged Cell Lines for the Study of Cell Cycle-Dependent Expression at the Single-Cell Level

Faithful measurement of dynamic changes in protein levels through the cell cycle based on fluorescence intensity requires that the tagged proteins of interest are expressed under the control of their endogenous regulatory sequences. This ensures that the dynamics of the labeled protein will recapitulate those of their native non-labeled counterpart. To achieve this, we generated stable double-tagged HeLa Kyoto cell lines using vectors generated via bacterial artificial chromosome (BAC) recombineering [24]. These vectors include the corresponding endogenous regulatory elements as well as exons and introns, which ensures splicing variant expression. Herein, we used a total of six such cell lines, which expressed EGPF-tagged ORC1, RIF1, Claspin, MCM6, or PCNA together with mCherry-tagged mPCNA. 

We measured the mean intensity for all fluorescently tagged proteins through the cycle (Appendix A). We focus on the mean rather than the total intensity because the former closely reflects differences in protein concentration. This concentration is a major determinant of the rate of reactions in which proteins participate. Therefore, combining fluorescence intensity data for a given protein of interest with data on its concentration in the same cell line would allow one to follow fluctuations in protein concentration during the cell cycle. We took advantage of published proteome-wide mass spectrometry-based measurements of protein concentrations for the HeLa Kyoto cell line [25], estimating the concentration profiles of our proteins of interest (Appendix A). It should be noted that the selected proteins function within the nucleus, and thus, the mean intensity allows us to overcome the increase in nuclear size through the course of the cell cycle. 

The average cell cycle length (measured as the period from the beginning of G1 to the end of M) of HeLa Kyoto cells under live-cell imaging was 15 h 58 min ± 1 h 56 min. The average duration of G1 was 5 h 25 min ± 1 h 13 min. On average, cells spent 7 h ± 49 min in S phase. The average duration of G2 was 2 h 38 min ± 45 min. M phase lasted 55 ± 34 min on average. Cells have variable cell cycle lengths, and the relative duration of cell cycle phases also varies. This does not allow for a direct comparison of fluorescence intensity between cells during the cycle. To overcome this, we performed normalizations for the whole cell cycle and its phases (Appendix A), which are described in detail in the Materials and Methods. As a result, we are able to calculate and compare mean intensity data for different proteins in the context of the cell cycle.

### 2.2. Sobel-Adjusted PCNA Fluorescence Intensity as an Indicator of Early, Middle, and Late S Phase

As is well-established, PCNA accumulates at replication foci [18,19,26,27], which are many and small at early S phase, becoming fewer and larger as S phase progresses. Since we used mPCNA-mCherry in the double-tagged cell lines, we imaged the human PCNA (hPCNA)-EGFP/mPCNA-mCherry line to confirm that the two orthologues exhibit identical changes in level and distribution through the course of the cell cycle (Figure 1D, Appendix A). Within a time-lapse, this phenotype allows us to discriminate between cell cycle phases [22]. Measurement of the mean PCNA intensity confirmed a gradual increase from the beginning of G1 until approximately 2 h after the appearance of the first replication foci, which was followed by a less pronounced increase until the end of S phase, followed by a plateau in G2 (Appendix A and Figure 1E). 

Expression changes alone are insufficient to effectively discriminate between early, middle, and late S phases. In light of the above-described differences in PCNA foci abundance and size throughout S phase, we sought to devise a method that can quantify this change in foci size and number, which would allow us to temporally define early, middle, and late S phases in an unbiased manner (i.e., without direct observation). To this end, we employed the Sobel operator, which is a gradient-based filter that amplifies the intensity at borders between structures (a brighter and a dimmer structure) in an image. Measurement of the intensity after applying the Sobel operator effectively discriminates between a nucleus containing a smaller and greater number of replication foci from their larger and less abundant counterparts. As a result, we clearly observed the beginning and end of S phase (Appendix A and Figure 1F). The former manifests as the start of a steep increase in Sobel-adjusted mean intensity, while the latter manifests as the end of a steep decrease (reflecting the dissolution of late replication foci). The initial steep increase is attributed to the rapid formation of a large number of small replication foci. This is followed by a plateau or decrease lasting approximately 2 h. We consider this period as the early stage of S phase. After early S, a continuous increase in Sobel-adjusted mean intensity takes place over a period of approximately 3.5 h (from 7.5 to 11 h). This is attributed to a considerable increase in foci size and intensity, which coincides with a decrease in their number, reflecting the middle S phase. In the following 1.5 h (from 11 to 12.5 h), the gradual disappearance of large replication foci marks the late S phase. Taken together, the measurement of Sobel-adjusted mean intensity allows us to faithfully discriminate between early, middle, and late subphases based on strict, reproducible, and quantifiable criteria. It should be noted that while we only employed the Sobel operator, other edge enhancement filters may also be successfully utilized for PCNA-based cell cycle phase discrimination.

### 2.3. RIF1

Utilizing this PCNA-based approach for cycle phase discrimination, we sought to follow multiple replication-associated proteins through the course of one cell cycle. We started with RIF1, a major regulator of the replication program from yeast to humans. It associates with chromatin at the beginning of G1 and accumulates at late-replicating origins. Multiple studies have shown that RIF1, through its interaction with PP1, represses origin firing by counteracting DDK-mediated helicase activation [28,29,30,31,32]. In light of its cell cycle-dependent function, we sought to characterize the RIF1 expression and localization throughout the cycle. Time-lapse imaging of the RIF1-EGFP/mCherry-PCNA double-tagged cell line revealed minor fluctuations in RIF1 mean fluorescence intensity and concentration throughout the cell cycle (Appendix A). Normalization of the mean fluorescence intensity between 0 and 1, which allows us to observe even minor dynamic changes in protein level, revealed an initial increase in RIF1 levels during G1. In fact, RIF1 levels peaked as early as 30 min into G1, followed by a steady decrease, which plateaued at the beginning of S (Figure 1A and Figure 2, and Appendix A). From the beginning of G1, RIF1 localizes at the nuclear periphery and around nucleoli, in line with its well-established recruitment to late-replicating regions [28,29,30,31,32]. Sobel-adjusted mean intensity corresponded to the mean intensity throughout the cell cycle, which suggested no significant changes in RIF1 distribution between phases (Figure 1G). Indeed, imaging revealed that the RIF1 localization pattern, established in early G1, remained constant thereafter, even during G2, after replication had ceased. Altogether, our results suggest that RIF1 levels peak early in G1 and considerably decrease until S, while its localization remains constant from G1 until M. 

### 2.4. MCM6

In light of the functional link between RIF1 and MCM6 in the replication program, we sought to determine and compare MCM6 levels and localization through the cell cycle [31]. MCM6 is a subunit of the MCM2-7 complex that serves as part of the replicative helicase in eukaryotes, which is loaded onto ORC-bound origins as part of the pre-replicative complex [33,34]. In contrast to RIF1, MCM6 mean fluorescence intensity and concentration fluctuated considerably through the cell cycle (Appendix A). Following normalization, MCM6 level dynamics closely resembled those of RIF1 throughout all phases, peaking in early G1 (~30 min) and progressively decreasing thereafter to reach a plateau throughout S and G2 (Figure 1B,F). During G1, approx. 100 min before the onset of S, traceable patterns of localization could be observed (~300 min), yet were not pronounced (Figure 3, Appendix A). MCM6 localization patterns became clearer as the cell progressed through G1. One possible explanation for this is that, as the cell progresses through G1, an abundance of free protein that “masks” foci is degraded. MCM6 accumulation was visible surrounding nucleoli, forming a considerably clearer pattern, which was most pronounced at the beginning of S. 

In contrast to RIF1 foci, which persisted during G2, the MCM6 pattern progressively disappeared through S phase. This is in line with the well-established notion that MCM6, as part of licensing machinery and the replicative helicase, is removed from chromatin when the fork passes through in order to avoid re-replication. The decrease in MCM6 protein levels throughout G1 and the parallel increase in MCM6 pattern, which reflects the chromatin-bound MCM6, suggest that the former can be attributed to a degradation of freely diffusing MCM6. In contrast to the mean MCM6 intensity measured herein, the total nuclear intensity of MCM4, another subunit of the MCM2-7 complex, was shown to gradually increase during S phase [35]. This can be attributed to an increase in nuclear size during the cell cycle (as the total nuclear intensity is being measured) or to a distinct expression pattern for MCM4. 

### 2.5. ORC1

Next, we followed cell cycle-dependent changes in the level and pattern of another licensing factor, namely, ORC1. ORC1 is the largest of six subunits that make up the origin recognition complex (ORC), which recognizes DNA at future replication origins in a sequence-independent manner in humans. Together with CDC6 and CDT1, ORC loads the MCM2-7 hexamer, thus assembling the pre-replicative complex [13,14]. Through bulk immunoblot-based analyses of synchronized cells, ORC1 was shown to be ubiquitinated and degraded in early S, then synthesized again during late G2 [36]. Our time-lapse imaging revealed that ORC1 concentration indeed fluctuated dramatically through the course of the cycle (Appendix A). Following a continuous increase in levels during G1, ORC1 reached a plateau at the beginning of S, which persisted for approximately 2 h. Entering middle S phase, ORC1 levels began to decrease, reaching a nadir at the beginning of G2 (Figure 1C and Figure 4, Appendix A). We attributed the observed decrease to the well-established degradation of ORC1. Time-lapse imaging of mCherry-tagged PCNA allowed us to pinpoint the onset of ORC1 decrease to the end of early S/beginning of middle S phase. Throughout the cell cycle, ORC1 exhibited a pattern rather than a homogenous distribution characteristic of a freely diffusing protein pool. This pattern transitioned from homogenous distribution throughout nuclear chromatin at the beginning of S to a preferential accumulation in the nuclear periphery during G2. As previously reported, we detected an increase in ORC1 levels toward the end of G2 and in mitosis [13,14,36,37,38,39,40]. In contrast to RIF1 and MCM6, ORC1 was associated with mitotic chromatin, as has been established [36]. 

### 2.6. Claspin

Claspin is an adaptor protein necessary for the ATR-mediated phosphorylation of Chk1 in response to replication stress [41]. Further, it maintains fork speed during unperturbed replication, with proposed roles in origin firing [42]. Claspin levels are subject to tight regulation during cell cycle progression and in response to checkpoint activation. Herein, we detected low Claspin levels throughout G1, followed by an increase that starts at the onset of S, reaching a plateau in the middle of S. This plateau persisted until entry into mitosis (Figure 1F and Figure 5, Appendix A). Mean fluorescence intensity-based decreases during mitosis may result from a dilution of the protein concentration following nuclear membrane disintegration when the same amount of (nuclear) protein ends up in the considerably bigger volume of a mitotic cell. As shown in Figure 1F, such a rapid decrease was observed for all proteins studied herein, with the exception of ORC1. In fact, ORC1 levels increase into M phase, which is due to its association with mitotic chromosomes, which are highly condensed relative to interphase chromatin. In line with our observations, synchronization coupled with immunoblot analysis revealed that Claspin levels are low during mitosis and G1, substantially increased during replication and G2 [40].

## 3. Materials and Methods

### 3.1. Cell Lines and Culture

mCherry-PCNA was introduced into HeLa Kyoto cell lines (RRID: CVCL_1922, sex: female) stably expressing fluorescently tagged RIF1, ORC1, PCNA, MCM6, Claspin, as previously described [24]. Cells were cultured in DMEM, high glucose, GlutaMAX™ (Thermo Fisher Scientific, Waltham, MA, USA) supplemented with 10% fetal bovine serum (FBS), 100 units/mL penicillin, and 100 μg/mL streptomycin at 37 °C and 5% CO_2_.

### 3.2. Image Acquisition

Time-lapse experiments were performed on a Nikon Eclipse Ti-E inverted microscope part of the Andor Revolution spinning-disk confocal system equipped with the Nikon Perfect Focus System (PFS). For all the acquired images, a Nikon CFI Plan Apo VC 60× (NA 1.2) water immersion objective (Nikon, Tokyo, Japan) and a high-sensitivity iXon897 Electron Multiplying Charge-Coupled Device (EMCCD) camera were used (Andor Technology – Oxford Instruments, Belfast, United Kingdom). The pixel size was 0.23 mm. For every time-lapse experiment, images were acquired on three Z planes with 0.2 µm spacing every 15 min. Two days prior to image acquisition, cells were plated in MatTek glass bottom dishes. On the day of imaging, the medium was changed to FluoroBrite DMEM (Thermo Fisher Scientific, Waltham, MA, USA) containing 10% FBS and 2 mM GlutaMAX Supplement (Thermo Fisher Scientific, Waltham, MA, USA). The Petri dishes were left to thermally equilibrate for 30 min before the start of image acquisition. During imaging, the cells were kept under optimal growth conditions at 37 °C and 5% CO_2_.

### 3.3. Image Analysis and Cell Cycle Normalization

For every time-lapse experiment, a maximum-intensity projection function was generated. Thereafter, the nucleus of every cell was selected as a region of interest (ROI) to obtain the mean intensity within the ROI, from which the background value (area between cells) was subtracted. Cells were followed from mitosis to mitosis as mitotic cells could be clearly distinguished as they have a characteristic spherical shape. Based on PCNA fluorescence (intensity and foci) [17,18,19,27], the specific cell cycle phase was determined. In Appendix A, we have provided the mean fluorescence intensity profiles of five cells with and without Sobel filtering, respectively. We normalized the cell cycle length of each cell by dividing each time point (such as 15, 30, 45 min, etc.) by the total cycle length for the given cell (such as 940 min), which results in a cycle length of 1 A.U. (Appendix A). After this normalization, the cycle lengths of all cells are equal to 1. However, the lengths of specific phases still varied (in other words, G1 is 0.31 A.U. in one cell and 0.27 A.U. in another). To make all phases equal between cells, we re-normalized each phase as done for the whole cell cycle, dividing each time point within the given phase by the length of that phase for the specific cell. At this point, the length of each phase was equal among the cells (Appendix A). However, to maintain the original proportion of each phase within the cycle, we multiplied each time point within a given phase by the average duration of this phase in all cells. As a result, the length of the cycle for all cells was the same (=1), while the proportion of each phase within the cycle remained identical to the average of all cells. This allowed us to determine the average fluorescence intensity of a given cell population over the cell cycle (Appendix A). 

Further, it “aligned” all cell cycles and respective phases for the measured cells, enabling the comparison of mean fluorescent intensities throughout the cycle. The cell cycle ranged from 0 to 1 A.U. After these normalizations, the intervals and, hence, time points differed between cells. To obtain intensity values at the same time points for all cells, we performed interpolation based on linear regression equations (FORECAST function in Microsoft Excel). Finally, we multiplied the value of each time point by the average cell cycle duration (15 h 58 min) in hours (Appendix A). This automatically gave us the average lengths of the phases, which indeed corresponded to the previously measured averages (Appendix A). 

For the Sobel operator values, we used the Sobel function in the CellTool software (version 1.6.0.7, available at https://dnarepair.bas.bg/software/CellTool/) [43], measuring the same ROI (cell nucleus) as for intensity. Using the same interpolation formula, we acquired the corresponding Sobel values. Data are presented as the mean and standard deviation.

### 3.4. Software 

For the measurement of intensity and Sobel values, we used CellTool software (version 1.6.0.7) [43]. Fiji software (version 1.54f) [44] was used for image preparation.

## 4. Conclusions

The current study of replication-associated factors through the cell cycle reveals a variety of dynamic profiles in terms of protein levels and nuclear distribution. In contrast to all other factors studied herein, RIF1 levels were relatively uniform throughout the cycle, with the exception of the M/G1 and G2/M transitions, which we attribute to nuclear envelope formation and breakdown, respectively. Further, RIF1 exhibited persistent localization to late-replicating regions during all phases except for M. While MCM6 exhibited the same normalized fluorescence intensity profile as that of RIF1 through the cycle, its concentration fluctuated 2–3-fold. Following a rapid increase in concentration during nucleus formation in early G1, MCM6 exhibited a gradual decrease until S phase. Meanwhile, the MCM6 pattern, which is a reflection of its binding to chromatin during pre-replication complex assembly, was visible an hour prior to the start of S phase. Rather counterintuitively, ORC1, which is required for MCM6 loading, exhibited a slower gradual increase in level throughout G1, followed by a rapid decrease until the end of S. Further, it exhibited chromatin enrichment as early as M phase. This enrichment was initially observed throughout the nucleus but then became exclusive to sites adjacent to the nuclear envelope in small amounts. Finally, we showed that Claspin levels increased from early S, reached a plateau in late S, and rapidly decreased at the G2/M transition. In the process, Claspin exhibited no specific pattern of distribution within the nucleus. Through the current approach, we successfully followed the concentrations of several DNA replication-associated proteins throughout phases of the cell cycle. This provides a practical basis for detailed modeling of the reactions that these proteins participated in.

## Figures and Tables

**Figure 1 ijms-25-08230-f001:**
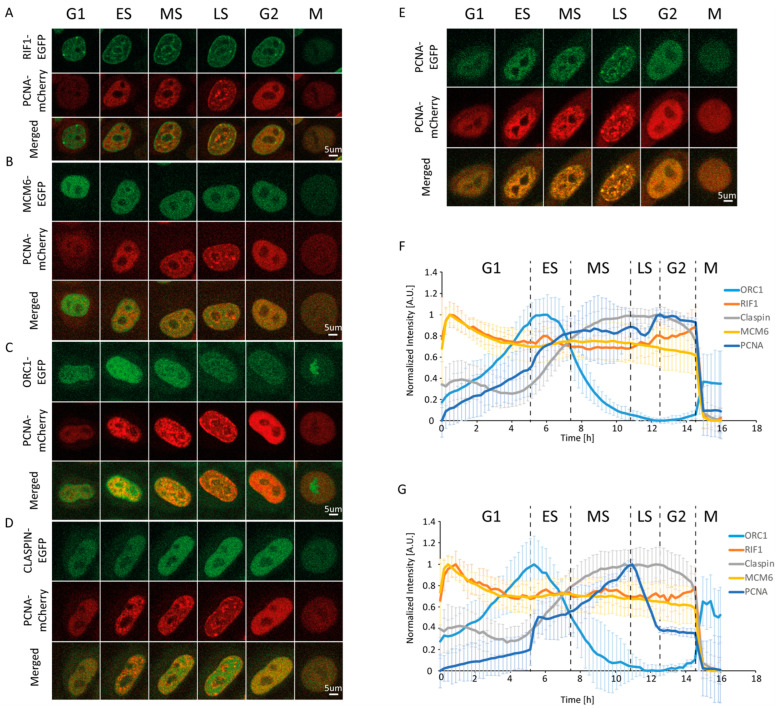
Changes in the levels and distribution of replication-associated proteins throughout the cell cycle. Representative time-lapse images of HeLa Kyoto cells co-expressing EGFP-tagged Rap1-interacting factor 1 (RIF1) (**A**), minichromosome maintenance complex component 6 (MCM6) (**B**), origin recognition complex subunit 1 (ORC1) (**C**), Claspin (**D**), or human proliferating cell nuclear antigen (PCNA) (**E**) together with mCherry-tagged mouse PCNA. (**F**) Normalized mean fluorescence intensity profiles of the studied proteins through the average cell cycle (15 h 58 min). (**G**) Profiles of normalized mean fluorescence intensity from (**F**) after Sobel filtering. Data are presented as mean ± SD.

**Figure 2 ijms-25-08230-f002:**
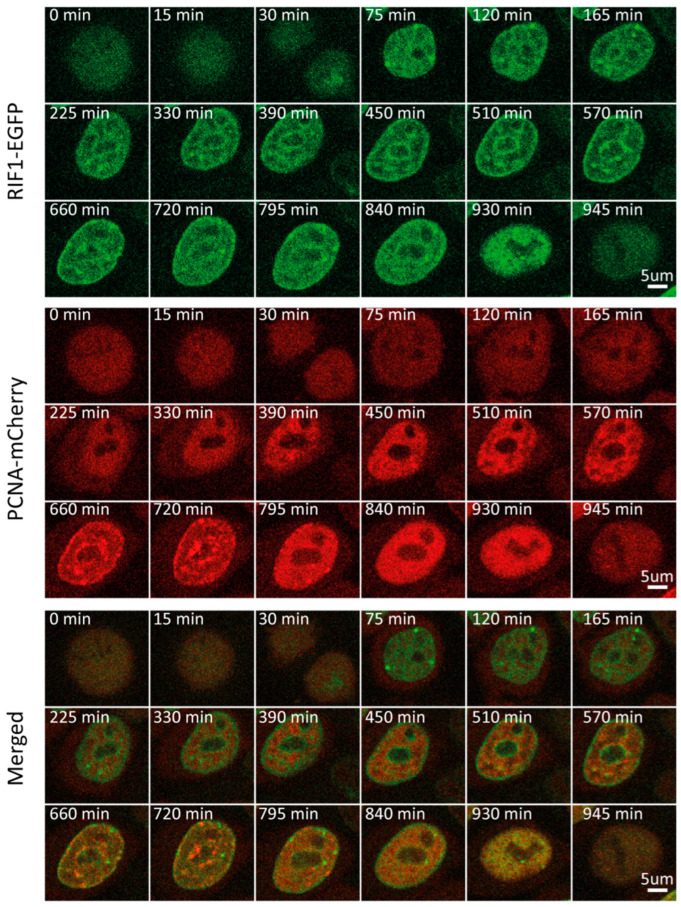
Representative time-lapse images of RIF1-EGFP/mCherry-PCNA HeLa Kyoto cells through the cell cycle.

**Figure 3 ijms-25-08230-f003:**
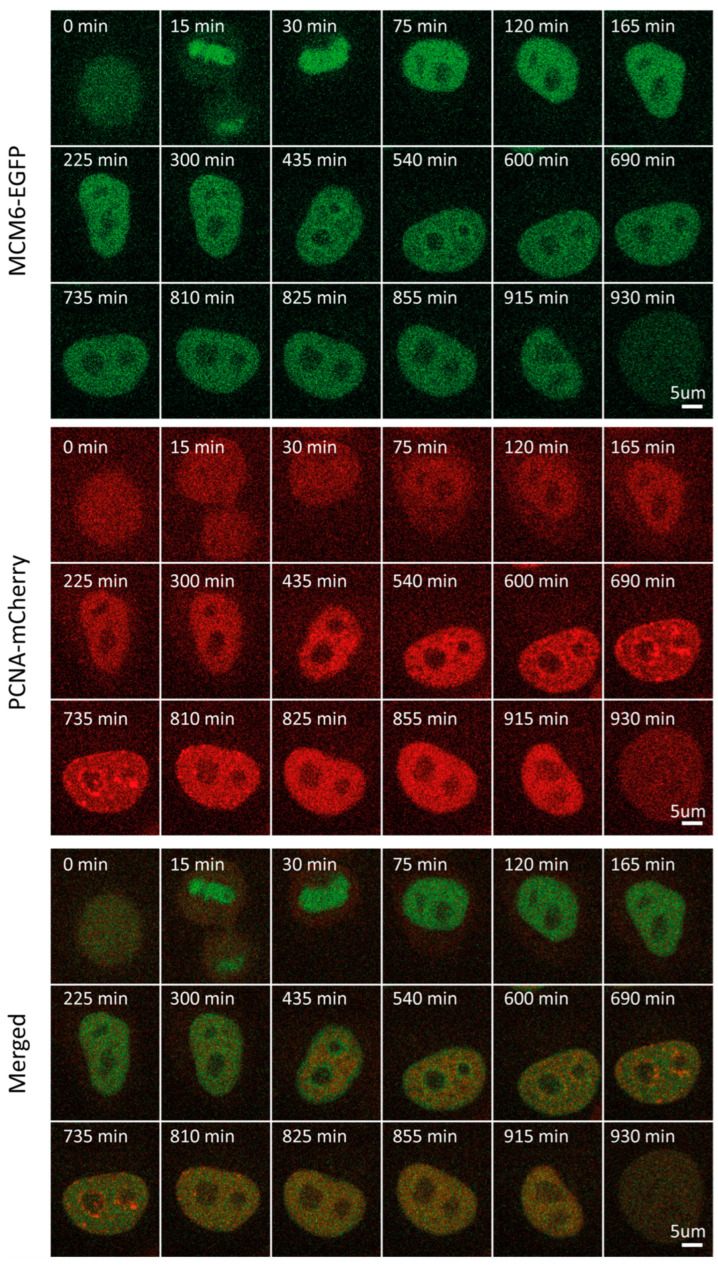
Representative time-lapse images of MCM6-EGFP/mCherry-PCNA HeLa Kyoto cells through the cell cycle.

**Figure 4 ijms-25-08230-f004:**
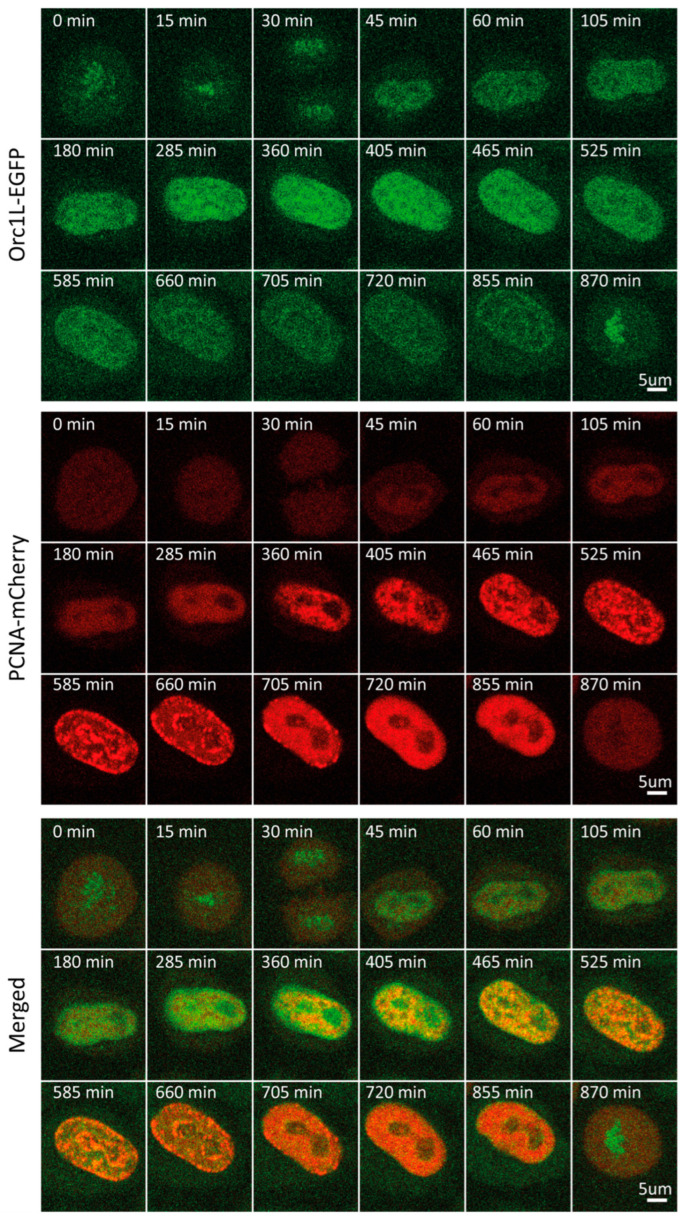
Representative time-lapse images of ORC1-EGFP/mCherry-PCNA HeLa Kyoto cells through the cell cycle.

**Figure 5 ijms-25-08230-f005:**
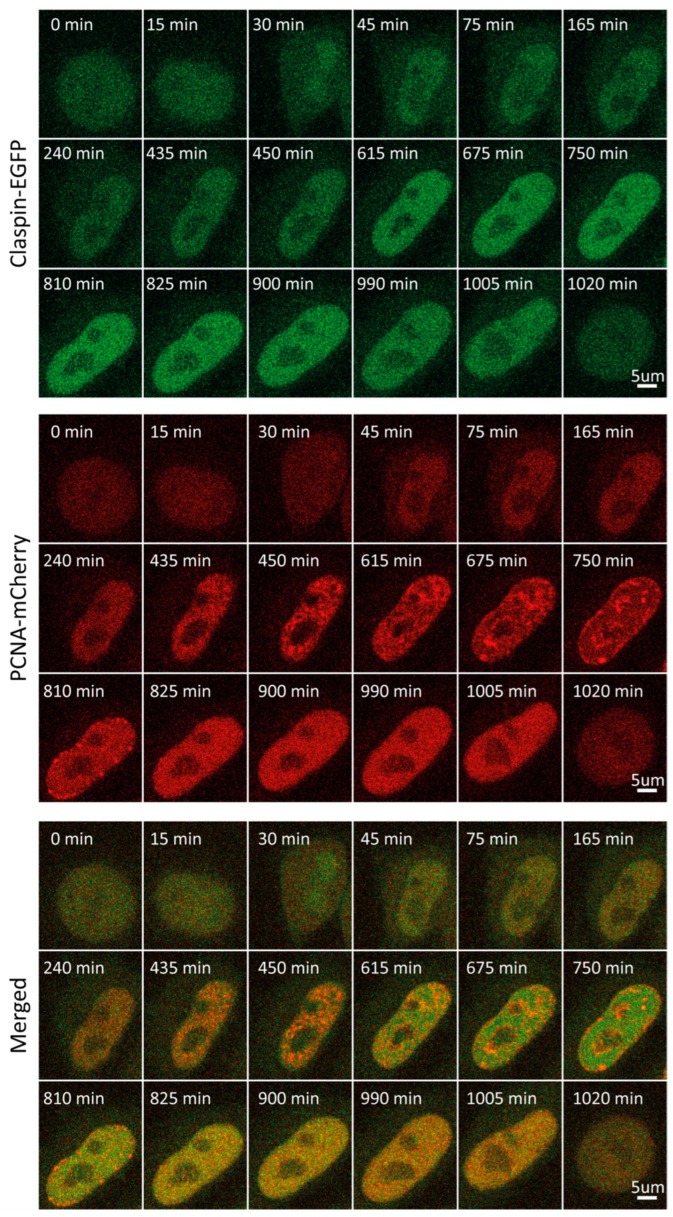
Representative time-lapse images of Claspin-EGFP/mCherry-PCNA HeLa Kyoto cells through the cell cycle.

## Data Availability

All data generated in this study are available from the corresponding author upon reasonable request.

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
