# Peer review of "Dynamics of Replication-Associated Protein Levels through the Cell Cycle"

_ijms, 2024, doi:10.3390/ijms25158230_

Round 1

Reviewer 1 Report

Comments and Suggestions for Authors

Atemin et al. have defined a new protocol for the identification and localization of proteins involved in the cell’s replication cycles by using fluorescence imaging approaches, and they applied their method to the analysis of single cells. The authors demonstrate how their methods provide insights into the spatiotemporal localization of proteins within single cells and report changes in concentration during different phases of the replication cycle.

The study is well-designed and suitable for publication; however, it lacks some additional information to corroborate their conclusions. 

  1. The authors should report the results of the application of the Sobel filters either in the results or in the supplemental materials. Additionally, the authors should justify the use of the Sobel filter and compare the results obtained by the application of the Sobel filter with those obtained using other edge enhancement filters, and clearly state the advantages of using the former.
  2. The dynamic profile of the proteins involved in the cell cycle is reported in the conclusions. However, it is unclear how these results compare with those previously reported in the literature. Can the authors clarify if the results are comparable with previous experiments? If not, can they highlight the differences and justify them?
  3. Have the authors performed any control experiments to determine whether the concentration and amount of proteins estimated from mass spectrometry data were accurately reflected in their experiment? If so, they should be included in the manuscript.

Reviewer 2 Report

Comments and Suggestions for Authors

This manuscript presents a methods to track the concentration of replication-associated proteins with spatiotemporal resolution. Using the double labeled HeLa cells, which express both PCNA-mCherry and EGFP-labeled replication-associated factors, the authors manage to measure the dynamic change of the replication-associated proteins concentration with a confocal microscope and proteome-wide mass spectrometry. Specifically, the PCNA-mCherry signal is used to distinguish the different stages of cell cycles, and the EGFP signal from various replication-associated proteins are used to determine the corresponding concentrations at each stage.

The manuscript organizes the experimental results into a coherent structure with a grammatical-free manner. The experimental results are sufficient to support the conclusion. More importantly, the imaging analysis method is detailed in this manuscript, which makes the experimental results more convincing.

The only suggestion I have is that the introduction of Sobel operator (in line 171 to 172) might be better to move to Introduction section, which familiarizes the audience with this method for following results.
